# Sesquiterpenes and Monoterpenes from the Leaves and Stems of *Illicium simonsii* and Their Antibacterial Activity

**DOI:** 10.3390/molecules27031115

**Published:** 2022-02-08

**Authors:** Huijuan Li, Xinghui Song, Huiru Li, Lifei Zhu, Shengbo Cao, Jifeng Liu

**Affiliations:** 1State Key Laboratory of Agricultural Microbiology, Huazhong Agricultural University, Wuhan 430070, China; lihj2022@126.com (H.L.); sxh811520@163.com (X.S.); 2School of Veterinary Medicine, Henan University of Animal Husbandry and Economy, Zhengzhou 450046, China; zhao_fx0309@163.com; 3Eastern China Conservation Centre for Wild Endangered Plant Resources, Shanghai Chenshan Botanical Garden, Shanghai 201602, China; wwwlihuiru@163.com; 4School of Pharmaceutical Science, Zhengzhou University, Zhengzhou 450001, China

**Keywords:** *Illicium simonsii*, sesquiterpene, monoterpene, antibacterial, MRSA

## Abstract

Two undescribed ether derivatives of sesquiterpenes, 1-ethoxycaryolane-1, 9*β*-diol (**1**) and 2-ethoxyclovane-2*β*, 9*α*-diol (**3**), and one new monoterpene glycoside, p-menthane-1*α*,2*α*,8-triol-4-O-*β*-D-glucoside (**5**), were obtained, together with eight known compounds from the stems and leaves of *I. simonsii*. Their structures were elucidated by spectroscopic methods. Compounds **1**–**11** were evaluated for their potency against *Staphylococcus aureus* and clinical methicillin-resistant *S. aureus* (MRSA). Among them, compound **3** was weakly active against *S. aureus* (MIC = 128 μg/mL), and compounds **6** and **7** exhibited good antibacterial activity against *S. aureus* and MRSA (MICs = 2–8 µg/mL). A primary mechanism study revealed that compounds **6** and **7** could kill bacteria by destroying bacterial cell membranes. Moreover, compounds **6** and **7** were not susceptible to drug resistance development.

## 1. Introduction

*Staphylococcus aureus* is one of the most devastating bacterial pathogens that can cause serious infections in both humans and animals [1]. The great challenge in the treatment of *S. aureus* infections is the appearance of methicillin-resistant *S. aureus* (MRSA) strains, which are not only developing resistance to conventional antibiotics, including β-lactams, macrolides, lincosamides, aminoglycosides and tetracyclines, but are also developing resistance to anti-MRSA antibiotics such as vancomycin, teicoplanin, linezolid and daptomycin [2,3,4]. There is an urgent need to discover and develop novel antibacterial compounds with therapeutic ability against these drug-resistant bacterial strains.

*Illicium simonsii* Maxim (Illiciaceae) is an evergreen tree or shrub and is distributed mainly in Southeast Asia [5]. Previous investigations on this plant have revealed that it is a rich source of essential oil, sesquiterpenoids, lignans and C_6_-C_3_ compounds [5,6,7].

According to our previous study, triphenyl-sesquineolignans isolated from *I. simonsii* exhibited good antibacterial activity against *S. aureus* and MRSA [8,9], which inspired us to further explore and confirm the antibacterial constituents of *I. simonsii*. In this paper, the isolation and structural elucidation of eleven terpenoids including three new compounds from *I. simonsii* were reported. In addition, their antibacterial activity was evaluated and the primary mechanism of action of the most active compounds, **6** and **7**, was further investigated.

## 2. Results

### 2.1. Chemistry

Compound **1**, a colorless gum, showed an absorption band for hydroxyl (3416 cm^−1^) in the IR experiment, and the HRESIMS data (*m*/*z* = 289.2144 [M+Na]^+^, calcd. for C_17_H_30_O_2_Na, 289.2144) indicate the molecular formula to be C_17_H_30_O_2_. The ^1^H NMR spectrum displayed characteristic signals due to three tertiary methyl groups (δ_H_ 0.98 (s, H_3_-13), 0.10 (s, H_3_-14) and 0.91 (s, H_3_-15)), and one ethoxy unit (δ_H_ 3.46–3.49 (2H, m, H-1′) and 1.14 (3H, t, *J* = 7.0 Hz, H_3_-2′)). The ^13^C NMR showed the presence of three tertiary methyl groups, five methylene groups, three methine groups (including one oxygenated carbon δ_C_ 72.6 (C-9)), three quaternary carbons (including one oxygenated carbon δ_C_ 75.3 (C-1)) and an ethoxy group (δ_C_ 57.3 and 16.0) (Table 1). The above NMR data of compound **1** are similar to those of caryolane-1, 9*β*-diol (**2**) [10], except the additional signals of an ethoxy unit and the downfield shift of C-1 (δ_C_ 75.3). These characteristic signals indicated that one ethoxy unit should be attached at C-1, which was further confirmed by the cross-peak between H-1′/C-1 in the HMBC spectrum (Figure 1). Compound **1** has the same relative configuration as compound **2** according to the ROESY experiment. Thus, the structure of compound **1** was elucidated as 1-ethoxycaryolane-1, 9*β*-diol.

Compound **3** was isolated as colorless gum, and its molecular formula was determined to be C_17_H_30_O_2_ based on HRESIMS (*m/z* = 289.2136 [M+Na]^+^, calcd. for C_17_H_30_O_2_Na, 289.2138). The IR spectrum of compound **3** showed the presence of a hydroxyl group at 3382 cm^−1^. Upon comparison of the ^1^H-NMR and ^13^C-NMR spectrum with those of clovane-2*β*, 9*α*-diol (**4**) [11], compound **3** exhibited additional signals of an ethoxy unit ((δ_C_ 65.8 and 15.7; δ_H_ 3.47–3.55 (2H, m) and 1.71 (3H, t, *J* = 7.0 Hz)) (Table 1). The ethoxy unit was found to be attached at C-2 by the observed downfield shift (δ_C_ 88.1, C-2) and further confirmed by the cross-peak between H-1′/C-2 in the HMBC spectrum (Figure 1). The relative configuration of **3** was the same as that of compound **4** according to the ROESY experiment. Thus, the structure of compound **3** was elucidated as 2-ethoxyclovane-2*β*, 9*α*-diol.

Compound **5** gave a pseudo-molecular ion peak at *m/z* 389.1744 [M+Na]^+^ (calcd. for C_16_H_30_O_9_Na, 389.1782) in the HRESIMS spectrum and exhibited a strong absorption band for hydroxyl groups (3416.6 cm^−1^) in the IR spectrum. The ^13^C NMR data revealed 16 carbon signals, consisting of a monoterpene skeleton and a D-glucopyranosyl moiety. The ^1^H NMR signals of anomeric protons of the D-glucopyranosyl moiety at δ_H_ 4.37 (1H, d, *J* = 7.7 Hz, H-1′) are consistent with the configuration for *β*-D-glucose. The NMR data of aglycone moiety of **5** were similar with those of (1*S*, 2*S*, 4*R*)-p-menthane-1,2,8-triol [12], except for the appearance of a quaternary carbon signal at δc 76.3 (C-4) derived from the connection with the glucose moiety (Table 2). This assignment was further confirmed by the HMBC correlations between H-1′ with C-4 (Figure 1). The observation of correlations H_3_-7/H-2, H-2/H_β_-3, H_β_-3/H-9 and Hα-3/H-1′ in the NOESY experiment confirmed the relative configuration of compound **5**, as shown in Figure 1. Thus, the structure of **5** was established as p-menthane-1*α*,2*α*,8-triol-4-O-*β*-D-glucoside.

Eight known compounds were identified as caryolane-1,9*β*-diol (**2**), clovane-2*α*,9*β*-diol (**4**) [10], pressafonin A (**6**), pressafonin B (**7**) [13], oplodiol (**8**) [14], ent-oplopanone (**9**) [15], abscisic acid (**10**) [16] and (3*S*,5*R*,6*S*,7*E*)-3,5,6-trihydroxy-7-megastigmen-9-one (**11**) [17] by comparison of their NMR spectroscopic data with those reported in the literature (Appendix A).

### 2.2. Antibacterial Activity

#### 2.2.1. Antimicrobial Activity Test of the *I. simonsii* Extract

The antimicrobial activity of the ethanol extract of *I. simonsii* was first determined for *S. aureus* ATCC 29213, *Bacillus. subtilis* ATCC 6633 and *Enterococcus faecalis* ATCC 29212. The zones of inhibition of 10 μg/disc of *I. simonsii* ethanol extract were 8.34 mm for *S. aureus*, 8.34 mm for *B. subtilis* and 7.70 mm for *E. faecalis* (Appendix A). It is evident that the *I. simonsii* extract might have active compounds responsible for the antimicrobial activity.

#### 2.2.2. Determination of Minimum Inhibitory Concentration (MIC)

The antimicrobial activity of compounds **1**–**11** was tested against both *S. aureus* (Gram-positive) and *Escherichia coli* (Gram-negative), and their MIC values are shown in Table 3. Compounds **6** and **7** exhibited good antibacterial activity against *S. aureus* with MIC = 4 and 8 μg/mL, respectively. Compound **3** was weakly active against *S. aureus* with MIC = 128 μg/mL. However, all compounds did not exhibit any activity against *E. coli* at tested concentrations lower than 128 μg/mL. The most active compounds, **6** and **7**, were further evaluated against 10 clinical MRSA strains using the same method. As shown in Table 4, the range of MIC values of compounds **6** and **7** was 2–8 μg/mL.

#### 2.2.3. Time−Kill Kinetics Assay

The time−kill kinetics assay of compounds **6** and **7** with different concentrations was carried out against MRSA-3. As shown in Figure 2 and Figure 3, both **6** and **7** showed rapid bactericidal activity. More specifically, the early exponential phase MRSA-3 could be completely eradicated by compound **6** (at 8 μg/mL (2× MIC) and 16 μg/mL (4× MIC); Figure 2a,c) and **7** (at 16 μg/mL (2× MIC) and 24 μg/mL (3× MIC); Figure 3a,c) within 1−2 h, and the growth of the stationary exponential phase MRSA-3 was also effectively prevented by compounds **6** (at 8 μg/mL and 16 μg/mL; Figure 2b,d) and **7** (at 16 μ*g*/mL and 24 μg/mL; Figure 3b,d), respectively.

#### 2.2.4. Resistance Development Study

The resistance development studies against *S. aureus* ATCC 29213 showed that compounds **6** and **7** were unsusceptible to drug resistance development even after 20 passages at subinhibitory concentrations (2 μg/mL and 4 μg/mL; 1/2× MIC, respectively), which demonstrated that it was difficult for *S. aureus* to develop resistance to both compounds, whereas norfloxacin showed an about 64-fold increase in the MIC after 14–16 passages (Figure 4).

#### 2.2.5. Hemolytic Activity

The preliminary toxicity of compounds **6** and **7** toward mammalian cells was evaluated against lyse red blood cells (RBCs). The 50% hemolysis concentration (HC_50_) of **6** and **7** were approximately 32 μg/mL (8× MIC) and 32 μg/mL (4× MIC), respectively (Figure 5).

### 2.3. Mode of Antimicrobial Action

#### 2.3.1. SYTOX Green Assay

In order to explore whether the bactericidal effect of active compounds acts directly on the cell membrane, compound **7** was selected to further investigate the mode of antimicrobial action. Experiments of the SYTOX green assay indicated that when an MRSA-3 bacterial suspension was exposed to 1×, 2×, 4× and 8× MIC of compound **7**, it increased the SYTOX green fluorescence emission gradually after 8 min (Figure 6a). In addition, when the concentration of compound **7** was 8× MIC, the fluorescence intensity was close to that of the positive control melittin (Figure 6b), while the blank, vancomycin and daptomycin failed to increase the fluorescence intensity under the same conditions. These observations confirmed that compound **7** rapidly disrupted bacterial membranes.

#### 2.3.2. Visualization of Bacterial Membrane Permeability

The influence of compound **7** on the MRSA-3 bacterial membranes was further visualized using 4′,6-diamidino-2-phenylindole (DAPI) and propidium iodide (PI), which are typically used to monitor the changes in bacterial viability [18]. As shown in Figure 7, the negative control without compound **7** only displayed blue fluorescence, indicating the intact cell membranes of MRSA-3, while after treatment with compound **7**, blue and red fluorescence was observed, suggesting that the cell membranes of MRSA-3 were disrupted by compound **7**.

## 3. Discussion

Several studies have illustrated that essential oils, terpenes and lignans from *Illicium* species, such as *I. verum*, *I. griffithii*, *I. wardii*, *I. angustisepalum* and *I. simonsii*, have potent antimicrobial activity [19,20,21,22,23]. To explain and confirm the antibacterial activity of *I. simonsii*, bioassay-guided fractionation of the ethanol extract of *I. simonsii* resulted in the isolation of three new natural compounds (**1**, **3** and **5**) and eight known compounds (Figure 1), in which compounds **3**, **6** and **7** are responsible for the antibacterial activity. The most active compounds (–)-bornyl p-coumarate (**6**) and (–)-bornyl cis-4-hydroxycinnamate (**7**) showed good activity against *S. aureus* and MRSA, with MIC values of 2–8 µg/mL. Similar structures exhibiting potent antibacterial activity were reported in previous studies, such as bornyl caffeate, bornyl coumarate [24], bornyl 3′, 4′-dimethoxybenzoate and bornyl 3′, 4′, 5′-tri-methoxybenzoate [25]. The antibacterial activity of compound **7** was slightly weaker than that of compound **6**, which may be attributed to the different configuration of double bonds in their structures. It is worth mentioning that (+)-bornyl p-coumarate previously isolated from *Piper caninum* also exhibited good antibacterial activity with MIC = 2 µM [24]. Perhaps the absolute configuration of their structure had no effect on the antibacterial activity. In order to further study the structure–activity relationship, we purchased the compounds (+)-borneol, (–)-borneol, p-coumaric acid and cis-4-hydroxycinnamic acid from Shanghai Aladdin Biochemical Technology Co., Ltd., and tested their antibacterial activity against *S. aureus*. However, they did not show activity against *S. aureus* (MICs > 128 μg/mL). Based on the above research, the esterified products of (±)-borneol and phenolic compounds can effectively improve their antibacterial activity; the position and the number of phenolic hydroxyl groups are closely related to their antibacterial activity [26]. These understandings could be helpful to propose a strategy for the development and rational design of more potent (±)-borneol ester-based antimicrobial derivatives.

In this study, the bacterial time–kill kinetics were assessed using clinical isolate MRSA-3 with vancomycin as the positive control. Compounds **6** and **7** could efficiently achieve a 4-log reduction at 4× MIC and 3× MIC both at the early exponential and stationary phases of cells within 1–2 h, respectively, whereas vancomycin needed more than 8 h to achieve this. The rapid antibacterial features of compounds **6** and **7** seem to indicate that their bactericidal mechanism may be achieved by damaging the bacterial cell membrane [27].

To verify our conjecture, the membrane disruption action of compound **7** against MRSA-3 was investigated by fluorescence probe SYTOX Green and PI, as well as fluorescence imaging technology. As a result, both assays presented the same results, showing that compound **7** could rapidly kill the bacteria by disrupting the cell membranes.

Antibiotic resistance is a highlighted problem that has attracted global attention. Many new and promising antibacterial candidates are being designed for targeting the bacterial cell membrane because of their low-frequency drug resistance and potential to rapidly kill bacteria [28], but their potential cytotoxicity to the host is still a matter of considerable debate [29]. In this case, the HC_50_ of compounds **6** and **7** against RBCs was approximately 32 μg/mL, which yielded an initial therapeutic index of 2–8 μg/mL towards MRSA. Despite this, the structural optimization of both compounds to reduce toxicity still needs to be further explored.

In conclusion, our present results detail three new terpenoids and two potent in vitro antibacterial constitutions in the stems and leaves of *I. simonsii* extract; moreover, compounds **6** and **7** may represent promising natural antibacterial compounds for developing new anti-infectious drugs, although their exact mechanism of action and pharmacological action in vivo remain to be clarified.

## 4. Material and Methods

### 4.1. General Experimental Procedures

Ultraviolet (UV) spectra and infrared (IR) spectra (KBr) were evaluated on a Shimadzu UV2401PC spectrophotometer (Shimadzu, Kyoto, Japan) and Bio-Rad FTS-135 spectrometer (Hercules, CA, USA), respectively. NMR spectra were obtained on DRX-400 or Advance III-600 spectrometers (Bruker, Bremerhaven, Germany) with TMS as the internal standard. Optical rotations were recorded on a JASCO P-1020 polarimeter (Horiba, Tokyo, Japan). The high-resolution mass spectra A were conducted on a Shimadzu LCMS-IT-TOF mass spectrometer (Shimadzu, Kyoto, Japan). Silica gel (100–200, 200–300 mesh, Qingdao Meigao Chemical Company, Qingdao, China), Sephadex LH-20 (20–150 μm, Amersham Pharmacia Biotech AB, Uppsala, Sweden.) and semi-preparative RP-HPLC purification (LC-20AT Shimadzu liquid chromatography system with ChromCoreTMC18 semi-preparative column, 250 mm × 10 mm, 5 μm) were used. Thin-layer chromatography (TLC) was conducted and detected under UV light or by heating after spraying with 10% H_2_SO_4_. Optical density and fluorescence were measured by a Microplate Reader (Tecan Infnite Pro series M200). A Nikon ECLIPSE 80i was used to perform the fluorescence experiments.

### 4.2. Plant Material, Separation and Purification of Compounds ***1***–***11***

The stems and leaves of the plant were collected in Dali City, Yunnan Province, China, during August 2015 and identified as *Illicium simonsii* Hook. f. et Thoms. by Prof. Dequan Zhang (College of Pharmacy, Dali University). A voucher specimen of *I. simonsii* (2 August 2015) was deposited at the School of Pharmaceutical Science, Zhengzhou University.

The powdered stems and leaves of *I. simonsii* (15.0 kg) were extracted with 95% EtOH (35 L × 2) under reflux for 3 h each time. After removal of solvent, the EtOH extract was suspended in H_2_O and then extracted with CH_2_Cl_2_ (5 L × 3) to give a CH_2_Cl_2_ fraction (300 g). The CH_2_Cl_2_ fraction was loaded onto a silica gel (CH_2_Cl_2_–MeOH, 100:1 to 80:20) and yielded seven fractions (Frs. 1–7). Fr. 4 (10 g) exhibited antibacterial activity with MIC = 128 µg/mL. Fr. 4 was further purified by repeated chromatography (including semi-preparative HPLC, silica gel, MCI gel, and Sephadex LH-20 gel) to obtain compounds **1** (4 mg), **2** (6 mg), **3** (15 mg), **4** (8 mg), **5** (16 mg), **6** (7 mg), **7** (15 mg), **8** (17 mg), **9** (23 mg), **10** (6 mg) and **11** (20 mg).

1-Ethoxycaryolane-1, 9*β*-diol (**1**). Colorless gum; [α${]}_{D}^{24.8}$+ 10.6 (*c* 0.180, MeOH); UV (MeOH): λ_max_ (log ɛ) = 196 (3.29) nm; IR (KBr): ν_max_ = 3416, 2931, 2867,1061, 1027 cm^–1^; ^1^H, ^13^C-NMR: Table 1; HR-ESI-MS: *m/z* = 289.2144 [M+Na]^+^ (calcd. for C_17_H_30_O_2_Na: 289.2144).

2-Ethoxyclovane-2*β*, 9*α*-diol (**3**). Colorless gum; [α${]}_{D}^{25.1}$ +5.42 (*c* 0.200, MeOH); UV (MeOH): λ_max_ (log ɛ) = 196 (3.23) nm; IR (KBr): ν_max_ = 3382, 2931, 1112, 1075 cm^–1^; ^1^H, ^13^C-NMR: Table 1; HR-ESI-MS: *m/z* = 289.2136 [M+Na]^+^ (calcd. for C_17_H_30_O_2_Na: 289.2138).

p-Menthane-1*α*,2*α*,8-triol-4-O-*β*-D-glucoside (**5**). Colorless gum; [α${]}_{D}^{19.1}$–16.26 (*c* 0.108, MeOH); UV (MeOH): λ_max_ (log ɛ) = 196 (3.63) nm; IR (KBr): ν_max_ = 3417, 2924, 1073, 1038 cm^–1^; ^1^H, ^13^C-NMR: Table 2; HR-ESI-MS: *m/z* = 389.1744 [M+Na]^+^ (calcd. for C_16_H_30_O_9_Na: 389.1789).

### 4.3. Antibacterial Assays

#### 4.3.1. Bacterial Strains and Growth Condition

*Staphylococcus aureus* (ATCC 29213), *Enterococcus faecalis* (ATCC 29212), *Bacillus subtilis* (ATCC 6633) and *Escherichia coli* (ATCC 25922) were purchased from the American Type Culture Collection. The methicillin-resistant *S. aureus* (MRSA-1–10) was provided by the First Affiliated Hospital of Zhengzhou University. The bacterial strains were stored in a refrigerator at –80 °C. Before the experiment, the experimental strains were thawed and inoculated on a MHA nutrient agar plate. After being cultured overnight at 37 °C, the strain was picked up into 1 mL MHB culture medium, and incubated in a 37 °C shaker at 200 rpm for 3–5 h.

#### 4.3.2. Antimicrobial Activity by K-B Disk Diffusion Test

The antimicrobial activity of the *I. simonsii* extract was carried out by the disc diffusion method according the literature [30]. Briefly, the test sample was first dissolved with DMSO to obtain a concentration of 1280 μg/mL solution, the inoculums of microorganisms were spread over nutrient agar plates with a sterile swab, and sterilized metrical filter paper discs were soaked with 10 μg/disc concentration of the test samples. Then, the soaked discs were placed on the marked agar plate and dried and inoculated at 37 °C for 16–18 h. Vancomycin was used as a positive control. The diameters of inhibition zones were measured.

#### 4.3.3. Minimum Inhibitory Concentration (MIC) Assay

The MIC values of compounds **1**–**11** were determined by broth micro-dilution in a Mueller–Hinton broth medium (MHB) according to the guidelines of the Clinical and Laboratory Standards Institute [31]. *S. aureus* (ATCC 29213), *E. coli* (ATCC 25922) and clinical MRSA strains (M-1–10) were used in this experiment. Test compounds were dissolved into DMSO, and antibiotic vancomycin was dissolved with sterile Milli-Q water, then these solutions were two-fold serially diluted to a series of gradient concentrations (256, 128, 64, 32, 16, 8, 4, 2, and 1 μg/mL). The results were determined after incubation for 16–18 h at 37 °C. The MIC was defined as the lowest concentration of the compound that produced complete inhibition of visible growth. All MIC determinations were repeated three times.

#### 4.3.4. Time–Kill Kinetics Assay

A time–kill assay was evaluated according to the guidelines of CLSI [31]. Briefly, an overnight culture of bacterial (MRSA-3) was adjusted in Mueller–Hinton broth (MHB) medium to obtain a bacterial suspension with 10^5^ (early exponential phase) and 10^8^ (stationary exponential phase) CFU/mL. Then, MASR-3 was challenged with compound **6** (1×, 2× and 4× MIC), **7** (1×, 2× and 3× MIC) and vancomycin (3× MIC, 4× MIC) at 37 °C and 225 rpm. The aliquots were removed at 0 h, 1 h, 2 h, 4 h, 6 h, and 8 h, serial 10-fold dilutions of the sample were made in the diluent (PBS), and a 10 μL aliquot of each dilution was plated out on MHA plates using the surface-spread plate method. The plates were incubated at 37 °C for 24 h to measure viable plate counts (CFU).

#### 4.3.5. Bacteria Resistance Study

According to the methods used in previous studies [32], the propensity for developing bacterial resistance towards *S. aureus* (ATCC 29213) of compounds **6**, **7** and control antibiotic norfloxacin was investigated. Briefly, the original MICs of compounds **6**, **7** and norfloxacin against *S. aureus* were determined at first. An *S. aureus* suspension incubated at the exponential phase at 37 °C was passaged to a fresh MHB containing the test compounds at sub-inhibitory concentration (1/2 MIC), the process was repeated every 20 to 22 h for up to 20 passages, and the MICs for test compounds were assayed at every passage as described above.

#### 4.3.6. Hemolytic Activity

Red blood cells (RBCs), isolated from the fresh sterile sheep blood, were resuspended in 1× PBS (5%). A RBCs suspension (150 μL) was then added to solutions of serially diluted compounds **6** and **7** at the different concentrations (compound **6** for 1/16×, 1/8×, 1/4×, 1/2×, 1×, 2×, 4×, 8× MIC; compound **7** for 1/4×, 1/2×, 1×, 2×, 4× MIC) in a 96-well plate (50 μL). Triton X-100 (1%) was used as a positive control. The plate was incubated for 1 h at 37 °C and centrifuged at 3500× *g* rpm for 5 min. The supernatant (100 μL) was transferred to a fresh 96-well plate. The results are shown as the percentage of hemolysis according to the absorbance measured at 540 nm [33].

### 4.4. Investigation of Antibacterial Mechanism

#### 4.4.1. SYTOX Green Staining Assay

The bacterial suspension of MRSA-3 was incubated at 37 °C for 6 h and suspended in 1× PBS. Then, the bacterial suspension was incubated with 3 μM of SYTOX green for 20 min at room temperature until the fluorescence readings were stable. The final concentrations of added compound **7** were 1/4× MIC, 1/2× MIC, 1× MIC, 2× MIC, 4× MIC, and 8× MIC. The fluorescence reading was monitored for the next 40 min at every 2 min interval at an excitation wavelength of 504 nm and an emission wavelength of 523 nm. Vancomycin and daptomycin were used as negative controls, and melittin was used as a positive control. DMSO (100%, 1%) and water were used as blank controls.

#### 4.4.2. Visualization of Bacterial Membrane Permeability

As described in the literature [34], the bacterial suspension of MRSA-3 was treated with compound **7** at desired concentrations (2× and 4× MIC). After incubation at 37 °C for 3 h, the bacterial suspension was added and stained with 4′, 6-diamidino-2-phenylindole dihydrochloride (DAPI, 10 μg/mL) and propidium iodide (PI, 20 μg/mL) for 30 min at 0 °C. Finally, the suspension was observed by a fluorescence microscope with an excitation wavelength of 485 nm. Bacterial no-treatment with **7** was used as a blank control.

## Figures and Tables

**Figure 1 molecules-27-01115-f001:**
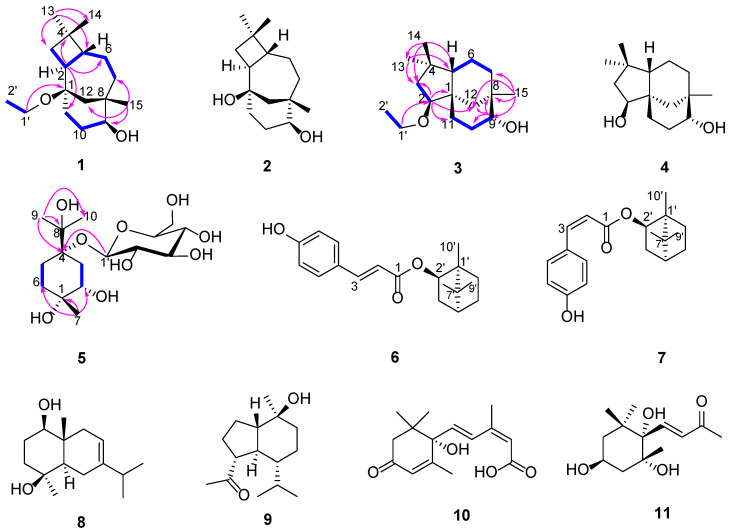
The structure of compounds **1**–**11**; Selected HMBC (**→**) and ^1^H-^1^H COSY (▬) correlations of **1**, **3** and **5**.

**Figure 2 molecules-27-01115-f002:**
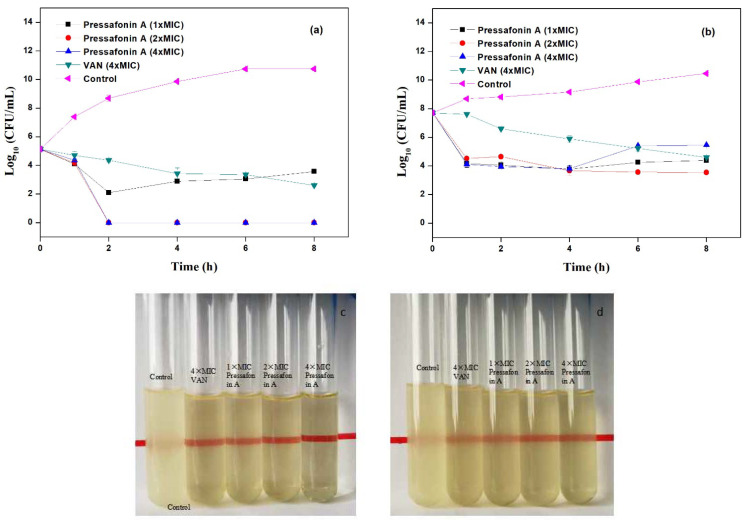
Time-dependent killing of MRSA-3 by compound **6**. (**a**,**c**) MASA-3 was grown to early exponential phase and was challenged with **6** (at 1×, 2× and 4× MIC) and vancomycin (at 4× MIC); (**b**,**d**) MASA-3 was grown to stationary exponential phase and was challenged with **6** (at 1×, 2× and 4× MIC) and vancomycin (at 4× MIC); (**c**,**d**) the bacterial suspension after treatment with compound **6** for 8 h. The control was treatment with sterile water.

**Figure 3 molecules-27-01115-f003:**
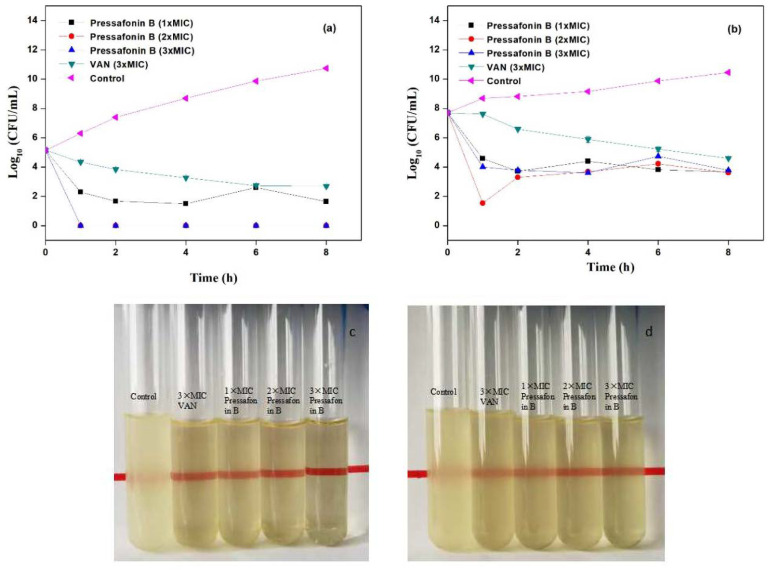
Time-dependent killing of MRSA-3 by compound **7**. (**a**,**c**) MASA-3 was grown to early exponential phase and was challenged with **7** (at 1×, 2× and 3× MIC) and vancomycin (at 3× MIC); (**b**,**d**) MASA-3 was grown to stationary exponential phase and was challenged with **7** (at 1×, 2× and 3× MIC) and vancomycin (at 3× MIC); (**c**,**d**) the bacterial suspension after treatment with compound **7** for 8 h. The control was treatment with sterile water.

**Figure 4 molecules-27-01115-f004:**
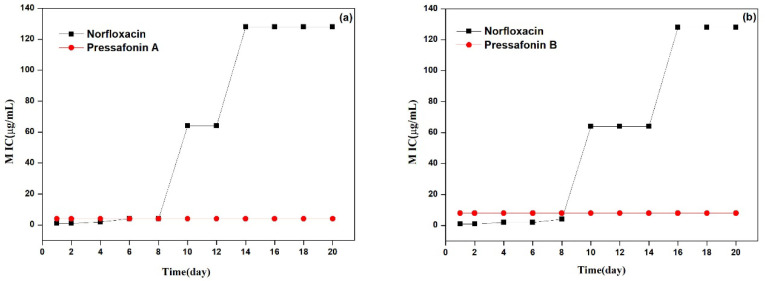
Bacterial resistance study of compounds **6** (**a**) and **7** (**b**) against *S. aureus* ATCC 29213 in the presence of 1/2× MIC levels. Norfloxacin (MIC = 2 μg/mL) was selected as a positive.

**Figure 5 molecules-27-01115-f005:**
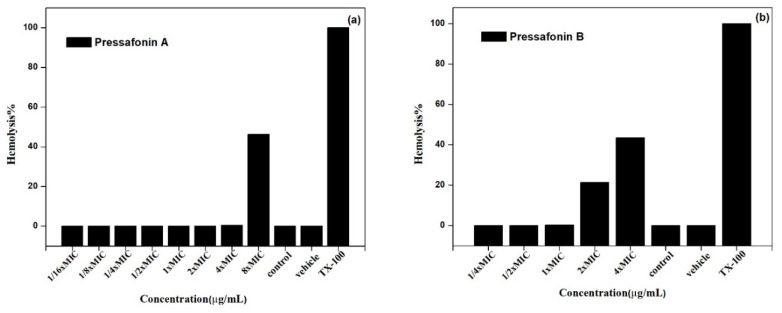
Percentage of hemolysis by **6** (**a**) and **7** (**b**) at different concentrations. Control: 50 μL of RBCs suspension (5%), vehicle: 1× PBS. All experiments were repeated three times.

**Figure 6 molecules-27-01115-f006:**
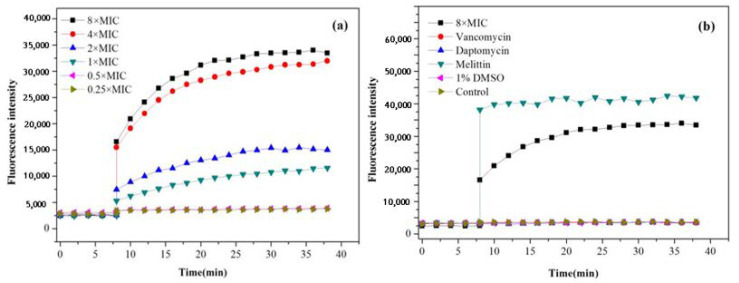
Mode of membrane disruption action of compound **7**. (**a**) Effects of compound **7** with different concentrations on the fluorescence intensity of SYTOX incubated with MRSA-3 at 1/4×, 1/2×, 1×, 2×, 4× and 8× MICs (MIC = 8 μg/mL); (**b**) effects of different antibiotics on the SYTOX fluorescence intensity; vancomycin and daptomycin were used as the negative control, and melittin was used as the positive control. The data represent mean values of three independent experiments.

**Figure 7 molecules-27-01115-f007:**
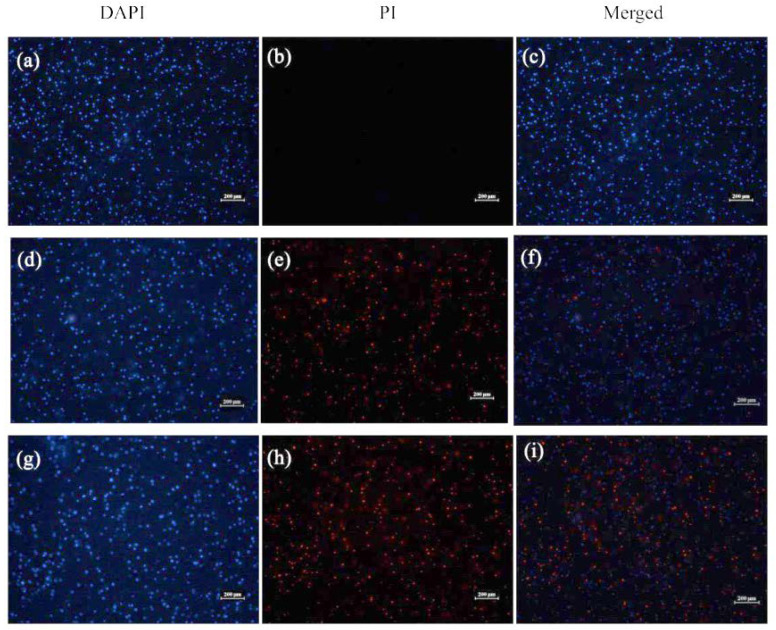
Fluorescence microscopy graphs of DAPI/PI staining of MRSA-3 treated with 2-fold and 4-fold MICs of **7**. (**a**–**c**) Non-treated with **7** (black control); (**d**–**f**) treated with **7** at 2× MIC; (**g**–**i**) treated with **7** at 4× MIC (scale bar 200 μm).

**Table 1 molecules-27-01115-t001:** ^1^H NMR and ^13^C NMR spectroscopic data of compounds **1** and **3** in CDCl_3_.

Position	1		3	
	*δ*_H_ (Multi, *J* in Hz)	*δ*_C_ (Multi)	*δ*_H_ (Multi, *J* in Hz)	*δ*_C_ (Multi)
1	–	75.3 (s)	–	44.2 (s)
2	2.09–2.14 (m)	38.8 (d)	3.42 (dd, 10.6, 5.5)	88.1 (d)
3*α*	1.61 (d, 9.3)	36.3 (t)	1.66 (dd, 11.8, 5.6)	45.0 (t)
3*β*	1.48–1.51 (overlap)		1.49 (dd, 11.4, 11.0)	
4	–	35.2 (s)	–	36.9 (s)
5	1.90–1.93 (m)	44.9 (d)	1.40–1.42 (overlap)	50.6 (d)
6*α*	1.51–1.53 (m)	20.9 (t)	1.40–1.42 (overlap)	20.6 (t)
6*β*	1.31–1.36 (m)		1.29–1.32 (m)	
7*α*	1.43–1.45 (m)	35.9 (t)	1.36–1.39 (m)	33.1 (t)
7*β*	1.12–1.16 (m)		1.09–1.11 (m)	
8	–	39.0 (s)	–	34.8 (s)
9	3.44–3.46 (m)	72.6 (d)	3.31 (br s)	75.3 (d)
10*α*	1.76–1.80 (m)	28.0 (t)	1.58–1.61 (overlap)	26.0 (t)
10*β*	1.96–2.01 (m)		1.95–2.01 (m)	
11*α*	1.73 (dd, 12.2, 5.5)	29.2 (t)	1.69 (dd, 13.7, 4.7)	26.7 (t)
11*β*	1.61–1.63 (m)		1.11–1.13 (m)	
12*α*	1.46–1.51 (overlap)	41.2 (t)	0.98–1.00 (m)	36.6 (t)
12*β*	1.46–1.51 (overlap)		1.58–1.61 (overlap)	
13	0.98 (s)	20.7 (q)	0.85 (s)	25.4 (q)
14	0.10 (s)	30.4 (q)	1.02 (s)	31.3 (q)
15	0.91 (s)	26.8 (q)	0.96 (s)	28.4 (q)
1′	3.46–3.49 (m)	57.3 (t)	3.47–3.55 (m)	65.8 (t)
2′	1.14 (t, 7.0)	16.0 (q)	1.71 (t, 7.0)	15.7 (q)

Data were recorded at 400 MHz for proton and at 100 MHz for carbon. Data assignments were based on HSQC and HMBC experiments.

**Table 2 molecules-27-01115-t002:** ^1^H NMR and ^13^C NMR spectroscopic data of compound **5** in CD_3_OD.

Position	5				
	*δ*_H_ (Multi, *J* in Hz)	*δ*_C_ (Multi)	Position	*δ*_H_ (Multi, *J* in Hz)	*δ*_C_ (Multi)
1	–	72.7 (s)	9	1.12 (s)	24.3 (q)
2	3.61–3.64 (m)	71.6 (d)	10	1.20 (s)	24.4 (q)
3*α*	1.64–1.67 (dd, 13.3, 3.7)	37.0 (t)	1′	4.37 (d, 7.7)	97.2 (d)
3*β*	2.36–2.40 (m)				
4	–	76.3 (s)	2′	3.01 (m)	73.8 (d)
5*α*	1.93–2.03 (overlap)	27.1 (t)	3′	3.21–3.24 (m)	76.9 (d)
5*β*	1.50–1.56 (m)				
6*α*	1.75–1.80 (m)	27.2 (t)	4′	3.15–3.18 (m)	70.2 (d)
6*β*	1.93–2.03 (overlap)				
7	0.94 (s)	22.4 (q)	5′	3.12–3.15 (m)	76.3 (d)
8	–	76.5 (s)	6′a	3.72–3.74 (dd, 12.0, 2.3)	61.4 (t)
			6′b	3.53–3.56 (dd, 11.9, 5.6)	

Data were recorded at 400 MHz for proton and at 100 MHz for carbon. Data assignments were based on HSQC and HMBC experiments.

**Table 3 molecules-27-01115-t003:** MIC (μg/mL) values of compounds **1**−**11** against *S. aureus* and *E. coli* in vitro.

	1	2	3	4	5	6	7	8	9	10	11	VAN ^c^	MER ^d^
*S. a* ^a^	>128	>128	128	>128	>128	4	8	>128	>128	>128	>128	2	ND ^e^
*E. c* ^b^	>128	>128	>128	>128	>128	>128	>128	>128	>128	>128	>128	ND ^e^	0.0625

^a^*S. a.*: *Staphylococcus aureus* ATCC 29213; ^b^
*E. c*: *Escherichia coli* ATCC 25922; ^c^ VAN: vancomycin; ^d^ MER: meropenem; ^e^ ND: not determined. Each test was repeated three times.

**Table 4 molecules-27-01115-t004:** MIC (μg/mL) values of compounds **6** and **7** against clinical MRSA strains in vitro.

Compound	Clinical Isolates of MRSA									
	M-1	M-2	M-3	M-4	M-5	M-6	M-7	M-8	M-9	M-10
**6**	2	4	4	2	4	2	4	4	4	4
**7**	4	8	8	4	8	4	8	8	4	8
Van ^a^	1	1	1	1	1	1	1	1	2	2

^a^ Van: vancomycin, positive control; The experiment was repeated three times.

## Data Availability

All data are available in this publication and in the Appendix A.

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
