# Peer review of "Sesquiterpenes and Monoterpenes from the Leaves and Stems of *Illicium simonsii* and Their Antibacterial Activity"

_molecules, 2022, doi:10.3390/molecules27031115_

Round 1

Reviewer 1 Report

Dear Authors,

I found this manuscript interesting, but I have several comments:

  • The name of compounds 6, 7 and 11 are not homogeneous in the text of the manuscript and in the supplementary material and can be confusing. Thus, in the text of the manuscript appear pressafonin A (6), pressafonin B (7) and respectively: (3S, 5R, 6S, 7E) -3,5,6-trihydroxy-7-megastigmen-9-one (11), and in the supplementary material: (-) - bornyl p-coumarate (6), (-) - bornyl cis-4-hydroxycinnamate (7) and 7-megastigmine-9-one (11), respectively. It would be better to use in all text only one choosed form of each compound. Also, in the Introduction section, lines 37-40 mention the following: "together with eight known compounds, caryolane-1,9β-diol (2) [4], clovane-2α, 9β-diol (4) [4], pressafonin A (6) [5], pressafonin B (7) [5], oplodiol (8) [6], ent-oplopanone (9) [7], abscisic acid (10) [8], (3S, 5R, 6S, 7E) -3,5,6-trihydroxy-7-megastigmine-9-one (11) [9] , have been isolated (Fig. 1). " The numbering in the round and square brackets are slightly confusing here. Specifically: for compounds: caryolane-1,9β-diol (2) [4], clovane-2α, 9β-diol (4) [4], in square brackets is the same specification for both compounds [4].
  • A very brief presentation of IR and UV spectroscopy data for each of the identified compounds may be useful in section 2.1 Chemistry.
  • I suggest that in the supplementary material to be included in English the data from the legend for S35 and UV spectrum of compounds 1, 3 and 5 

Author Response

I found this manuscript interesting, but I have several comments:

  • The name of compounds 6, 7 and 11 are not homogeneous in the text of the manuscript and in the supplementary material and can be confusing. Thus, in the text of the manuscript appear pressafonin A (6), pressafonin B (7) and respectively: (3S, 5R, 6S, 7E) -3,5,6-trihydroxy-7-megastigmen-9-one (11), and in the supplementary material: (-) - bornyl p-coumarate (6), (-) - bornyl cis-4-hydroxycinnamate (7) and 7-megastigmine-9-one (11), respectively. It would be better to use in all text only one choosed form of each compound.

Response: The names of compounds 6, 7 and 11 have been modified in the supplementary material.

  • Also, in the Introduction section, lines 37-40 mention the following: "together with eight known compounds, caryolane-1,9β-diol (2) [4], clovane-2α, 9β-diol (4) [4], pressafonin A (6) [5], pressafonin B (7) [5], oplodiol (8) [6], ent-oplopanone (9) [7], abscisic acid (10) [8], (3S, 5R, 6S, 7E) -3,5,6-trihydroxy-7-megastigmine-9-one (11) [9], have been isolated (Fig. 1). " The numbering in the round and square brackets are slightly confusing here. Specifically: for compounds: caryolane-1,9β-diol (2) [4], clovane-2α, 9β-diol (4) [4], in square brackets is the same specification for both compounds [4].

Response: Thank you for your suggestion. We have modified the relevant sentences

  • A very brief presentation of IR and UV spectroscopy data for each of the identified compounds may be useful in section 2.1 Chemistry.

Response: There are obvious hydroxyl characteristic peaks in the IR spectrum, however, no obvious characteristic signals in the UV spectrum. The manuscript has been modified according to your suggestions.

  • I suggest that in the supplementary material to be included in English the data from the legend for S35 and UV spectrum of compounds 1, 3 and 5 

Response: Thank you for your suggestion. We have translated the important parameters of the IR and UV spectrum into English.

Reviewer 2 Report

  1. The manuscript needs though the grammar checker and sentence construction
  2. The manuscript needs rewriting abstract and introduction lines 12,13 and in abstract copy from lines  34, 35, 36, and 37 from the introduction. results not written in the introduction.
  3. Line 188, in the discussion section, The 80 % ethanol extract of the stems and leaves of I. simonsii inhibited the growth of tested strains S. aureus ATCC29213 (8.34 mm), Bacillus. subtilis ATCC6633 (8.34 mm), and E. faecalis ATCC29212 (7.70 mm) at 10 μg/disc in K-B disk diffusion test. Where are these results? These results were not recorded in the results section.
  4.  Authors mention Bacillus subtilis (ATCC6633) line 266 without being used in any test.
  5. Line 236 Plant material and isolation of compounds 1–11 correct into Plant material, separation, and purification of compounds
  1. Lines 246,247,248 contains results that must be deleted
  2. From line 252 into 263 results must be deleted and added to the results section.
  3. Line 319  The bacterial solution must be correct into bacterial suspension.
  4. The manuscript needs rewriting discussion and discuss all results from the separation of a compound, compound elucidated by spectroscopic, antibacterial activity, and Mode of antimicrobial action. Finally, add more references to the discussion.

Author Response

  1. The manuscript needs though the grammar checker and sentence construction

Response: We have polished the grammar and sentences as much as possible in the revised manuscript.

  1. The manuscript needs rewriting abstract and introduction lines 12,13 and in abstract copy from lines  34, 35, 36, and 37 from the introduction. results not written in the introduction.

Response: Thank you very much for your suggestion. We have rewritten these paragraphs

  1. Line 188, in the discussion section, The 80 % ethanol extract of the stems and leaves of I. simonsii inhibited the growth of tested strains S. aureus ATCC29213 (8.34 mm), Bacillus. subtilis ATCC6633 (8.34 mm), and E. faecalis ATCC29212 (7.70 mm) at 10 μg/disc in K-B disk diffusion test. Where are these results? These results were not recorded in the results section.

Response: These experimental results have been placed in supplementary materials

  1.  Authors mention Bacillus subtilis (ATCC6633) line 266 without being used in any test.

Response: Bacillus subtilis (ATCC6633) was used in the activity test of ethanol extract of Illcium simonsii.

  1. Line 236 Plant material and isolation of compounds 1–11 correct into Plant material, separation, and purification of compounds

Response: Thank you for your suggestion, the manuscript has been modified.

  1. Lines 246,247,248 contains results that must be deleted

Response: Thank you for your suggestion, the contains results has been deleted.

  1. From line 252 into 263 results must be deleted and added to the results section.

Response: Thank you for your suggestion, the contains results has been deleted.

  1. Line 319 The bacterial solution must be correct into bacterial suspension.

Response: Thank you for your suggestion, The bacterial solution has been correct into bacterial suspension.

  1. The manuscript needs rewriting discussion and discuss all results from the separation of a compound, compound elucidated by spectroscopic, antibacterial activity, and Mode of antimicrobial action. Finally, add more references to the discussion.

Response: Thank you very much for your suggestion. We have revised the discussion section. Chemical section, we discussed it in detail. The primary antibacterial mechanism of active compounds has been carried out. Therefore, only the results were discussed.

Round 2

Reviewer 2 Report

Dear / Author

After greeting

Although the author, do some of the comments, there are still some comments needed to correct as the following:-

  1. Figure 1 combined with Figure 2 to become figure 1 and transfer to results.
  2. Line 138, in the results section, However, all compounds showed no activity against coli (MIC > 128 μg/mL). correct into However, all compounds do not exhibit any activity against E. coli at concertation more than (128 μg/ml).
  3. Discussion, it still needs more discussing about all results obtained until becoming more fit.
  4. Line 221, 222, 223 in the discussion section. The 80 95 % ethanol extract of the stems and leaves of simonsii inhibited the growth of tested strains S. aureus ATCC29213 (8.34 mm), Bacillus. subtilis ATCC6633 (8.34 mm), and E. faecalis ATCC29212 (7.70 mm) at 10 μg/disc in K-B disk diffusion test (Supporting information). These results must be added in the result section and E. faecalis ATCC 29212 to materials and methods section line 330.
  5. Take space between ATCC and code number aureus ATCC29213 corrected into S. aureus ATCC 29213 on manuscript text.

Moreover,  the manuscript still needed more revision.

Thank you

Prof. Gamal M. El-Sherbiny

Author Response

  1. Figure 1 combined with Figure 2 to become figure 1 and transfer to results.

Response: We have revised the Figure 1 according to your suggestion

  1. Line 138, in the results section, However, all compounds showed no activity against coli (MIC > 128 μg/mL). correct into However, all compounds do not exhibit any activity against E. coli at concertation more than (128 μg/ml).

Response: Due to the limitation of solvent concentration, the highest concentration we tested was 128 μg/ml. In the revised manuscript, we correct it into “However, all compounds do not exhibit any activity against E. coli at the tested concertation lower than 128 μg/mL.”

  1. Discussion, it still needs more discussing about all results obtained until becoming more fit.

Response: According to your suggestion, we analyzed and discussed our experimental data more detail.

  1. Line 221, 222, 223 in the discussion section. The 80 95 % ethanol extract of the stems and leaves of simonsii inhibited the growth of tested strains S. aureus ATCC29213 (8.34 mm), Bacillus. subtilis ATCC6633 (8.34 mm), and E. faecalis ATCC29212 (7.70 mm) at 10 μg/disc in K-B disk diffusion test (Supporting information). These results must be added in the result section and E. faecalis ATCC 29212 to materials and methods section line 330.

Response: Thank you for your suggestion. We have revised it in the manuscript.

  1. Take space between ATCC and code number aureus ATCC29213 corrected into S. aureus ATCC 29213 on manuscript text.

Response: Thank you for your suggestion. We have revised it in the manuscript.

Moreover,  the manuscript still needed more revision.

Response: We have revised the manuscript, and the revised sentences are marked in red color.